# The Protective Role of Hydrogen Sulfide Against Obesity-Associated Cellular Stress in Blood Glucose Regulation

**DOI:** 10.3390/antiox9111038

**Published:** 2020-10-23

**Authors:** Ania Mezouari, Radhika Nangia, Jeffrey Gagnon

**Affiliations:** Department of Biology, Laurentian University, Sudbury, ON P3E 2C6, Canada; amezouari@laurentian.ca (A.M.); rnangia23@gmail.com (R.N.)

**Keywords:** hydrogen sulfide, antioxidant, glucoregulation, GLP-1

## Abstract

Circulating palmitic acid (PA) is increased in obesity and causes metabolic stress, leading to diabetes. This includes the impairment of the glucoregulatory hormone glucagon-like peptide-1 (GLP-1) secreted from intestinal L-cells. Recently, the anti-inflammatory gasotransmitter hydrogen sulfide (H_2_S) has been implicated in the enhancement of GLP-1 secretion. We hypothesized that H_2_S can reduce the oxidative stress caused by palmitate and play a protective role in L-cell function. This study was conducted on both human and mouse L-cells and a mouse model of Western diet (WD)-induced obesity. PA-induced L-cell stress was assessed using DCF-DA. H_2_S was delivered using the donor GYY4137. C57BL/6 mice were fed either chow diet or PA-enriched WD for 20 weeks with ongoing measurements of glycemia and GLP-1 secretion. In both L-cell models, we demonstrated that PA caused an increase in reactive oxygen species (ROS). This ROS induction was partially blocked by the H_2_S administration. In mice, the WD elevated body weight in both sexes and elevated fasting blood glucose and lipid peroxidation in males. Additionally, a single GYY4137 injection improved oral glucose tolerance in WD-fed male mice and also enhanced glucose-stimulated GLP-1 release. To conclude, H_2_S reduces oxidative stress in GLP-1 cells and can improve glucose clearance in mice.

## 1. Introduction

Obesity has reached epidemic proportions worldwide with a prevalence that has tripled since 2016, and a projection that nearly half of the US population will be obese by 2030, as reported by the World Health Organization. Obesity is a disease defined based on the calculation of the body mass index (BMI) being higher than 30 kg/m^2^ and, more specifically, with an abnormal fat depot. Excessive body fat accumulation in adipocytes is an established cause of metabolic impairment [1] and is closely linked with complications such as heart disease, stroke, and type 2 diabetes mellitus [2].

Medications that successfully combat obesity are limited. One group of anti-obesity treatments that are being used are the Glucagon-like peptide-1 (GLP-1) drugs [3]. These either prolong the half-life of endogenous GLP-1 such as dipeptidyl peptidase-4 (DPP4) inhibitors, or act as GLP-1 receptor agonists. The latter successfully target obesity (1–3 kg after 26/30 weeks), whereas Dipeptidyl peptidase-4 inhibitors are weight—neutral [4,5]. GLP-1 increases insulin release, enhances insulin action, acts as a satiety signal, and slows gastric emptying, leading to reduced food intake and weight loss [6]. Normally, GLP-1 is secreted by endocrine L-cell found in the colon and the ileum in response to food consumption [7,8]. Unfortunately, GLP-1 secretion may be impaired in obese individuals, and this may contribute to the hyperglycemia and decreased satiety seen during obesity [9]. The mechanism leading to obesity-induced GLP-1 impairment is not clearly known.

As fat accumulation correlates with systemic oxidative stress [10], and this can lead to impaired function of cells [11,12], these effects may explain the impaired GLP-1 secretion seen during obesity. In particular, elevated palmitic acid (PA, 16:0) has been implicated in metabolic stress through increases in oxidative stress and elevation of cellular reactive oxygen species (ROS) [13]. Elevated ROS will contribute to cellular dysfunction if not balanced by antioxidants.

Hydrogen sulfide (H_2_S) is a gasotransmitter that acts as a signaling molecule and as an antioxidant [14]. It is produced endogenously from L-cysteine by the enzymes cystathionine gamma-lyase and cystathionine beta-synthase [15], and by the gut microbiome by a variety of sulfate-reducing bacteria (SRB) [16]. Importantly, these SRB are localized in the distal ileum and colon, which is the same niche as GLP-1 secreting L-cells [7]. Recently, we demonstrated a role for bacterial H_2_S in the stimulation of GLP-1 secretion [17]. However, the precise mechanism of how this occurs and whether H_2_S can protect from obesity-induced metabolic stress is not known. As a follow up to this work, we set out to examine the role of H_2_S as a protective antioxidant under palmitate-induced metabolic stress. This was examined in GLP-1 secreting L-cells and obese mice.

## 2. Materials and Methods 

### 2.1. Cell Culture

The human NCI-H716 cell line was obtained from American Type Culture Collection (ATCC). The mouse GLUTag cell line was obtained from Dr. D. Drucker (Lunenfeld-Tanenbaum Research Institute, Toronto, ON, Canada). Cell lines were maintained in 10 cm plates containing Roswell Park Memorial Institute (RPMI 1640, GE Healthcare Life Sciences, South Logan, UT, USA) medium for the NCI-H716 cells and low-glucose Dulbecco’s Modified Eagle’s Medium (DMEM, Sigma Aldrich, Oakville, ON, Canada) for GLUTag. Both mediums were supplemented with 1% penicillin-streptomycin (Pen Strep; Life Technologies, South Logan, UT, USA), and 10% fetal bovine serum (FBS) (Life Technologies, Waltham, MA, USA) and incubated at 37 °C in 5% CO_2_. Experiments were performed on twenty-four-well plates, which were seeded with cells at a density of 200,000 cells per well for NCI-H716 cells and 250,000 cells per well for GLUTag cells and allowed to proliferate for 48 h. Matrigel (Corning Life Sciences, Bedford, MA, USA) was used to secure the NCI-H716 cells to a basement membrane matrix [18].

### 2.2. PA/FAF-BSA Conjugation Preparation

This study uses the fatty acid preparation method of Cousin [19]. Stock 5mM PA/fatty acid-free bovine serum albumin (FAF-BSA) was prepared as follows. 100 mM PA (Sigma Aldrich, Saint Louis, MO, USA) stock solution was dissolved in 0.1 M NaOH and heated at 70 °C in a water bath. Simultaneously, a 10% (wt/vol) fatty acid-free FAF-BSA (EMD Millipore Corp, Billerica, MA, USA) solution was prepared in deionized H_2_O in a 55 °C water bath. To prepare a 5 mM PA/10% FAF-BSA stock solution, 0.5 mL of the 100 mM PA was added to 9.5 mL 10% BSA solution, which was then heated at 55 °C in a shaking water bath for 10 min before it was vortex-mixed for 10 s and then incubated for an additional 10 min in a 55 °C water bath. The PA/FAF-BSA complex solution was cooled to room temperature before being sterile filtered using a 0.45 μm pore size filter. Control for this complex was prepared the same way by using just regular BSA (GE Healthcare Life Sciences, GE Healthcare Life Sciences, South Logan, UT, U.S.A). The prepared 5 mM PA/FAF-BSA and regular BSA aliquoted solutions were stored at −20°C, where they were stable for three weeks. The stored PA/FAF-BSA and its control stock solutions were heated for 15 min at 55 °C and cooled to room temperature before the study.

### 2.3. Intracellular ROS Detection

Levels of intracellular ROS were measured using the cell-permeable dye, 2′,7-Dichlorofluorescein diacetate (DCF-DA, Sigma-Aldrich, Oakville, ON, Canada). DCF-DA is oxidized in the presence of ROS to produce Dichlorofluorescin (DCF), which is a highly fluorescent compound. After 48 h, the cells were washed with phenol red free Hank’s Balanced Salt Solution (HBSS, Sigma-Aldrich, Oakville, ON, Canada) and incubated in 1 mL of HBSS (1x) with 100 μM DCF-DA for 10 min and 30 min for the NCI-H716 and GLUTag cells, respectively. As DCF-DA is sensitive to light and air, the plates were covered with aluminum foil at all times. 

The stock 5 mM PA/FAF-BSA was diluted in HBSS to make various treatment concentrations. Regular BSA stock solutions were added to HBSS to match PA concentrations for control. After the 10-min or 30-min dye incubation, the cells were washed with HBSS, and the treatments and controls were added. The cells were then returned to the incubator. Fluorescence was measured using the Fluostar Optima plate reader (BMG Labtech, Offenburg, Germany) at 15 min then at 30-min intervals for two hours. These readings were measured with the excitation filter set to 485 nm and emission filter at 520 nm. 

### 2.4. Cytotoxicity Assessment of Palmitate

To check for the cytotoxic effects of PA, a Neutral Red assay was conducted using the protocol outlined by Repetto [20] with slight modifications. 0.020 mL of Neutral Red dye (Sigma-Aldrich, Oakville, ON, Canada) was diluted in 5 mL of each treatment before 1 mL of the treatment was added to each well. The cells were then returned to the incubator. After 2 h, treatments and dye were aspirated before cells were washed with a wash fixative, composed of distilled water with 1% calcium chloride and 0.37% (vol/vol) formaldehyde (Sigma-Aldrich, Oakville, ON, Canada). An extraction solution composed of distilled water with 50% (vol/vol) bonded ethanol and glacial acetic acid 1% (vol/vol) (Sigma-Aldrich, Oakville, ON, Canada) was then added. Cells were gently agitated on a Brunswick TM Innova ® 2100 Platform Shaker (Eppendorf Canada, Mississauga, ON, Canada) for 5 min to allow for the complete extraction of cells. Readings were measured with absorbance set to 600 nm.

### 2.5. Animals

5 or 6 weeks old C57BL/6 mice of both sexes were purchased from Charles River Laboratories (St. Constant, Quebec) and kept in the Paul Field Animal Care Facility at Laurentian University. Animals were housed in groups on standard 12:12-h light:dark cycles. The Laurentian University Animal Care Committee approved protocols following guidelines of the Canadian Council for Animal Care (CCAC, Ottawa, ON: Vol. 1, 2nd edition, 1993: Vol. 2, 1984).

### 2.6. In Vivo Experimental Design

Upon arrival in the facility, 15 female and 15 male mice were randomly divided into two groups according to diets. Mice had access to water and food *ad libitum* and were either fed a Western diet (n = 11) (Research Diets, Inc. New Brunswick, NJ, USA, 43% Carbohydrates, 40% fats: 31.32% PA, 17% proteins and 14.5% salt (energy percent)) or a chow diet (n = 4) (Envigo Teklad, Madison WI, 40% Carbohydrates, 5.5% fats: 0.7% PA, 22% proteins and 0.4% salt (energy percent)) *ad libitum* for 20 weeks. Blood was collected every two weeks after an oral glucose gavage (OGTT) for glucose and GLP-1 measurements. 

### 2.7. Oral Glucose Tolerance Test and GLP-1 Determination

An oral glucose tolerance test (OGTT) was conducted at week 16 of the study. Mice were first injected intraperitoneally with 30 mg/kg GYY4137 or a vehicle then immediately were fasted overnight for 16 h before receiving oral gavage of D-(+)-glucose (2 g/kg body wt). Blood glucose was measured from the lateral saphenous vein during the experiment using a glucometer (OneTouch Verio) at 0, 5, 30, 60, and 90 min after glucose administration. For GLP-1 measurement, blood was collected from the lateral saphenous vein into EDTA coated capillary tubes (Sarsdedt) at 0, 5, and 30 min after glucose gavage. Diprotin A (DPP-4 inhibitor) and aprotinin (a protease inhibitor) were added to the capillary tubes (10% *v*/*v*) to prevent degradation of the target hormone. The capillary tubes were centrifuged at 5000× *g* for 5 min at 4 °C. 10 μL of plasma was examined for total GLP-1 using a commercial competitive ELISA kit (Millipore). 

### 2.8. In Vivo Oxidative Stress Analysis

Levels of circulating lipid peroxidation were examined to assess oxidative stress in the Western diet and chow diet mice plasma (20 μL) of both sexes. The end product during lipid peroxidation, malondialdehyde (MDA), was measured in blood plasma in a 96-well plate with a colorimetric assay kit (Abcam, Cambridge, MA, USA). Plasma MDA reacts with thiobarbituric acid (TBA), producing the MDA-TBA complex that is quantified at 532 nm using a microplate absorbance reader.

### 2.9. Statistical Analysis

All values were presented as mean ± standard error of mean. A one-way ANOVA test analyzed data with multiple doses, followed by a Bonferroni post hoc test. A two-way ANOVA test analyzed data with two or more independent variables, followed by Bonferroni post hoc test at each time point. A Student’s *t*-test was used to analyze experiments comparing two groups. Statistical significance was considered at *p* < 0.05.

## 3. Results

### 3.1. Palmitate Stimulates ROS Production in Both NCI-H716 and GLUTag Cells

First, we developed a protocol for ROS detection and examined the effect of PA on ROS production in NCI-H716 and GLUTag cell lines. In NCI-H716 cells, PA increased ROS in a time and dose-dependent manner. The 500 μM PA, relative to its control, caused the most significant stimulation of ROS by threefold at 120 min (3.655 ± 0.186-fold of control, *p* < 0.0001, Figure 1a). In the GLUTag cell line, all the doses tested showed significant stimulation of ROS compared to the corresponding BSA controls by at least one-fold at 120 min (2.074 ± 0.183-fold of control, *p* < 0.000, Figure 1b). To determine cytotoxicity when cells were treated with treatments of PA, cells was assessed via neutral red assay. Relative to BSA controls, the 500 μM of palmitate was not toxic to NCI-H716 (Figure 1c). However, a modest but significant effect was observed in GLUTag cells (0.915± 0.090-fold of control, *p* < 0.05, Figure 1d). 

### 3.2. GYY4137, a Slow-Releasing H_2_S Donor, Reduces Basal and Palmitate-Induced ROS in NCI-H716 Cells

To investigate the direct effect of H_2_S on ROS, NCI-H716 cells were treated with a slow-releasing H_2_S donor, GYY4137. GYY4137 at 1mM significantly reduced basal ROS by half after 120 min of treatment (0.593 ± 0.022-fold of control, *p* < 0.0001, Figure 2A). 500 μM PA-induced ROS was significantly reduced by 25% by co-incubating cells with 1 mM of GYY4137 (1.219 ± 0.036-fold with GYY4137 versus 1.478 ± 0.036-fold without GYY4137, *p* < 0.0001, Figure 2B).

### 3.3. Western Diet Elevates Body Weight in Both Sexes and Elevates Fasting Blood Glucose and Lipid Peroxidation in Male Mice

To investigate the effect of elevated PA in vivo, mice were fed either a chow diet or a Western diet rich in PA. Western diet elevated body weight in both sexes with a pronounced weight difference in males at 16 weeks (43.445 ± 0.761 g for WD-fed mice compared to 33.050 ± 1.485 g for chow diet-fed mice, *p* < 0.0001, Figure 3B) that appeared much earlier than in the females (31.145 ± 1.896 g for WD-fed mice versus 23.950 ± 0.866 for chow diet-fed mice, *p* < 0.0001, Figure 3A). No significant elevation in fasting blood glucose was seen in females at 16 weeks (7.775 ± 0.795 mM for WD-fed mice versus 6.475 ± 0.226 mM for chow diet-fed mice, Figure 3C) while a much greater elevation was seen in males (10.550 ± 0.572 mM for WD-fed mice versus 7.900 ± 0.593 mM for chow diet-fed mice, *p* < 0.0001, Figure 3D). Circulating lipid peroxidation was assessed by MDA assay and levels were only elevated in the WD-fed males (184.515 ± 2.011 nmol/mL for WD-fed mice versus 112.660 ± 1.845 nmol/mL for chow diet-fed mice, *p* < 0.0001, Figure 3E).

### 3.4. The H_2_S Donor, GYY4137, Improves Oral Glucose Tolerance in Western Diet-Fed Male Mice

Blood glucose response after an oral glucose challenge (2 g/kg) was examined in fasted mice given an IP injection of GYY4137 (30 mg/kg) or saline at −16 h. No significant change was observed in females injected with H_2_S compared to control (Figure 4A,B). Glucose tolerance was significantly improved in males injected with the H_2_S donor (*p* < 0.01 for treatment effect, Figure 4C,D). 

### 3.5. The H_2_S Donor, GYY4137, Enhances Glucose Stimulated GLP-1 Release in Western Diet-Fed Mice

Plasma GLP-1 response to an oral glucose challenge was examined in WD-fed mice given an intraperitoneal (IP) injection of GYY4137 (30 mg/kg) or saline at −16 h. Animals (combined male and female) receiving GYY4137 had a greater elevation (delta) in 5 min GLP-1 release (12.774 ± 4.402 for GYY4137-treated mice compared to 2.041 ± 3.205 for control, *p* < 0.05, Figure 5A). Absolute GLP-1 at time 0 (basal) was similar in between groups (25.7 pM in control vs. 27.3 pM in GYY4137). The enhancement in GLP-1 was more pronounced in male mice as WD-fed males had completely lost their GLP-1 response (peak of 16.728 ± 7.015 for GYY4137-treated mice compared to −0.626 ± 5.991 for control, *p* < 0.05, Figure 5C). Males had a similar basal GLP-1 concentration of 28.4 pM in control and 30.3 pM in GYY4137. In females, a much smaller improvement was observed (peak at 6.843 ± 1.673 for GYY4137-treated mice compared to 4.709 ± 2.602 for control, Figure 5B). Basal GLP-1 levels were also similar in females with 24.1 pM in control and 25.8 pM in GYY4137 treated. 

## 4. Discussion

Obesity is associated with elevated circulating free fatty acids including PA, a highly abundant saturated fatty acid found in high-fat diets [21]. The excessive dietary amounts of PA have been implicated in metabolic stress resulting from oxidative stress elevation by constant ROS production [13]. Oxidative stress causes cytotoxicity that eventually results in cell dysfunction, leading to apoptosis when those cells show a low antioxidant defense and cannot counteract the excess amount of ROS production [11]. Therefore, PA-induced oxidative stress may explain GLP-1 secretion impairment during obesity. While current studies recognize the cytotoxicity that may arise from elevated levels of PA in vitro in chronic conditions [22,23], they do not determine to what extent this cytotoxicity impairs L-cells, and whether this oxidative stress can be reversed. 

To explore the effects of acute PA treatment in physiological and pathophysiological simulations, three doses of PA—125 μM, 250 μM, and 500 μM—were used. To mimic circulating plasma PA during obesity, lipotoxic conditions can be simulated by incubating cells with 500 μM of palmitate [23]. Indeed, we found the 500 μM PA caused the most significant production of ROS by over threefold in NCI-H716 cells and twofold in GLUTag cells. This is in agreement with previous work in GLUTag’s demonstrating PA-induced ER stress [23]. Interestingly, in our work the NCI-H716 cells required the 500 μM dose of PA to induce ROS, whereas the GLUTag’s had significant induction of ROS at all the tested doses. We also note, that the degree of ROS induction in the NCI-H716 cells was nearly double that of the GLUTag cells. Although NCI-H716 and GLUTag cell lines are both enteroendocrine L-cell models, they vary considerably from one another, and this may explain the differences seen in dose-dependent ROS generation [24]. One possible explanation is that the levels of endogenous antioxidants may vary between the two cell lines. There may be higher levels of antioxidants present in the NCI-H716 cells, which can counteract the lower does PA-induced levels of ROS. Indeed, when the cells were examined for PA-induced cytotoxicity using the neutral red test, a significant reduction in viability was observed in GLUTag and not NCI-H716 cells. Additional experiments exploring the antioxidant capacity of each of these cell models may resolve this discrepancy. It is important to recognize that in vivo, a variety of fatty acids and cell stress inducers would be present and that an animal model of high fat diet-induced oxidative stress will better recapitulate the fatty acid environment of L-cells.

While we did not explore the precise mechanism of palmitate’s ROS inducing effects in L-cells, this saturated fatty acid is a well-established inducer of ER stress, Superoxide anion generation, and altered cellular calcium regulation, reviewed in [13]. Recently, palmitate’s ability to impair GLP-1 secretion in L-cells (induced by insulin) has been described [23]. In this work, L-cells exposed to palmitate had altered processing of the GLP-1 precursor, proglucagon, which led to glucagon production rather than GLP-1. The reduced GLP-1 and enhanced glucagon production, are likely major contributors to the lipotoxicity-induced impaired glucose regulation. This known lipotoxic effect led us to explore a protective mechanism. 

In the cell experiments, we demonstrated the antioxidant capacity of the H_2_S donor GYY4137. On its own, and in PA-induced ROS generation, H_2_S was able to reduce ROS levels. H_2_S is well established as an antioxidant gas. Hydrogen sulfide exhibits antioxidant proprieties in two ways. First, H_2_S has a weak reducing propriety, which can react chemically with the superoxide anion [25]. Second, H_2_S can upregulate antioxidant enzymes such as glutathione peroxidase [26,27] and superoxide dismutase [28] through its S-sulfhydration action [29,30]. Future in-depth cell studies will elucidate the mechanism being employed in L-cells.

Next, we explored the effects of the PA-rich Western diet (WD) oxidative stress, GLP-1 regulation, and glucose regulation in C57BL/6 mice. As expected, WD elevated body weight in both female and male mice [31]. However, males had a greater increase in body weight at an earlier point in the study and exhibited a greater elevation in fasting blood glucose relative to females. Importantly these indications of metabolic dysfunction aligned with the degree of circulating peroxidated lipids found in males (measured as MDA levels). While MDA does not provide a clear sense of the oxidative stress within the L-cells, it does serve as a proxy for the degree of oxidative stress in the animal [32]. It is possible the females were less sensitive to WD-induced metabolic dysfunction due to the elevated levels of estrogen. Estrogen is known to induce lipid oxidation for use as an energy source [33] and also plays a beneficial role in glucose homeostasis, as low estrogens can lead to insulin resistance [34]. Since our experimental design included both male and female mice, we were able to detect this difference in WD sensitivity.

We then tested the effect of a single injection of the H_2_S donor GYY4137 in WD-fed mice. While we were not equipped to measure the precise blood concentration of H_2_S before and 16 h after GYY4137 delivery, we selected GYY4137 over sulfide salts as concentrations remain 3 days once in solution [35]. With a 30 µg/g GYY4137 IP dose, and based on a mouse blood volume estimates, H_2_S levels in the blood were likely in the µM range. Future spectroscopic measurement of H_2_S and sulfur derivatives including persulfides and polysulfides in blood will further clarify this.

GYY4137 caused a significant improvement in glucose tolerance and GLP-1 secretion relative to control. It’s important to note that female mice, despite being on the WD for 16 weeks, did not exhibit the same degree of impaired glucose tolerance or impaired GLP-1 secretion compared to males. Indeed, the GLP-1 response to oral glucose was completely lost in WD-fed male mice. It should be noted, that despite the significant overall effect of GYY4137 on GLP-1 response in male and female combined data, neither sex separately reached statistical significance. To determine whether H_2_S has the ability to rescue WD-induced metabolic impairment in females, future studies may require a longer diet duration relative to the one used in males. It should be noted that our route of administration for the H_2_S donor was IP. When injecting the compound, H_2_S will diffuse throughout the body [26] including the intestinal villi [36] where L-cells are located. Indeed, IP injections of GYY4137 have been previously shown to impact villus structure [36] suggesting the compounds ability to reach the intestine.

The enhanced GLP-1 secretion in mice receiving GYY4137 is in line with our labs’ previous work in both L-cells and in mice [16]. In that study, GYY4137 enhanced GLP-1 secretion by nearly twofold compared with vehicle-treated GLUTag cells. In mice, H_2_S levels were increased by administering a probiotic diet that enhanced sulfate-reducing bacteria. While this previous work provided a novel mechanism using the gut microbiome to enhance H_2_S and GLP-1, the current study demonstrated a role for H_2_S to protect from fatty acid-induced metabolic stress. Indeed, the potential role of H_2_S as a therapeutic treatment is promising as trials are underway exploring its anti-inflammatory properties [37].

## 5. Conclusions

In conclusion, our work demonstrates that H_2_S not only reduces oxidative stress induced by PA in L-cells but also enhances GLP-1 secretion and improves glucose clearance in mice. This study will lay the foundation for future work exploring H_2_S as a potential therapeutic agent in the treatment of complications associated with obesity. 

## Figures and Tables

**Figure 1 antioxidants-09-01038-f001:**
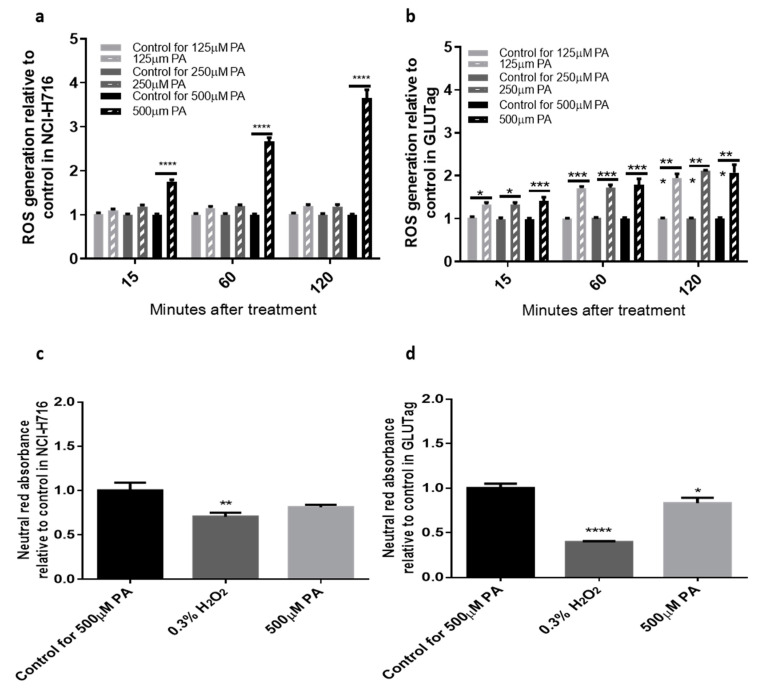
Palmitate-induced reactive oxygen species (ROS) production in NCI-H716 and GLUTag cells. ROS generation (**a**,**b**) was determined after a 2-h incubation with 2′,7′-DCF-DA and palmitate treatment and regular bovine serum albumin (BSA) as control. Cell viability (**c**,**d**) was assessed via neutral red absorbance after a 2-h incubation with neutral red dye and indicated treatments. Data were analyzed using a Two-way ANOVA followed by Bonferroni post hoc test, * = *p* < 0.1, ** = *p* < 0.01, *** =*p* < 0.001, **** =*p* < 0.0001. Results are expressed as means ± SEM, n = 8.

**Figure 2 antioxidants-09-01038-f002:**
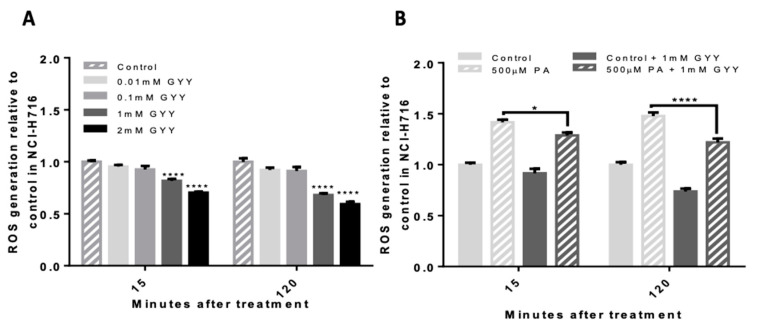
The antioxidant effect of H_2_S donor on basal and palmitate-induced ROS in NCI-H716 cells. Fluorescence was analyzed after a 2-h incubation with 2′,7′-DCF-DA and treatments. (**A**) 1 mM and 2 mM of GYY4137 significantly reduces basal ROS by nearly half. (**B**) 500 μM PA-induced ROS was significantly reduced by co-incubating cells with GYY4137 at 1 mM. Data were analyzed using a Two-way ANOVA followed by Bonferroni post hoc test, * = *p* < 0.01, **** = *p* < 0.0001). Results are expressed as means ± SEM n = 8.

**Figure 3 antioxidants-09-01038-f003:**
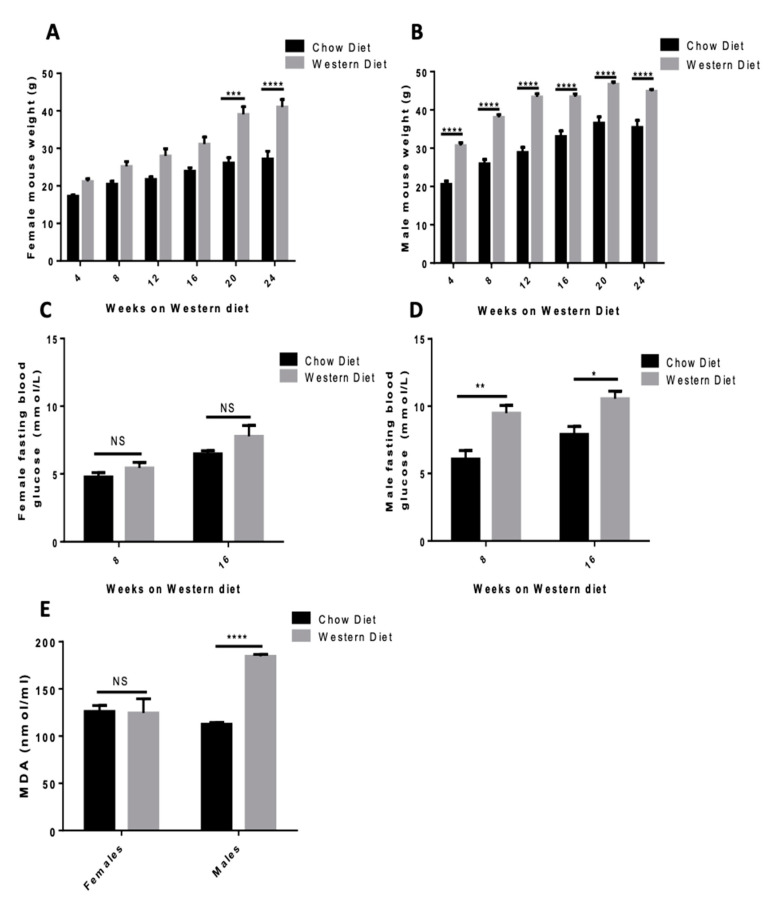
Western diet effect on mice body weight, fasting blood glucose, and lipid peroxidation. Body weight was examined in female (**A**) and male (**B**) mice maintained for 24 weeks on a Western or chow diet. Fasting blood glucose at 8 and 16 weeks was examined in female (**C**) and male mice (**D**) on a Western and chow diet. (**E**) Circulating lipid peroxidation was examined in both sexes after 24 weeks on a Western or chow diet. Data were analyzed using a Two-way ANOVA followed by Bonferroni post hoc test, * = *p* < 0.1, ** = *p* < 0.01, *** =*p* < 0.001, **** = *p* < 0.0001). Results are expressed as means ± SEM n = 11 Western diet, n = 4 chow diet.

**Figure 4 antioxidants-09-01038-f004:**
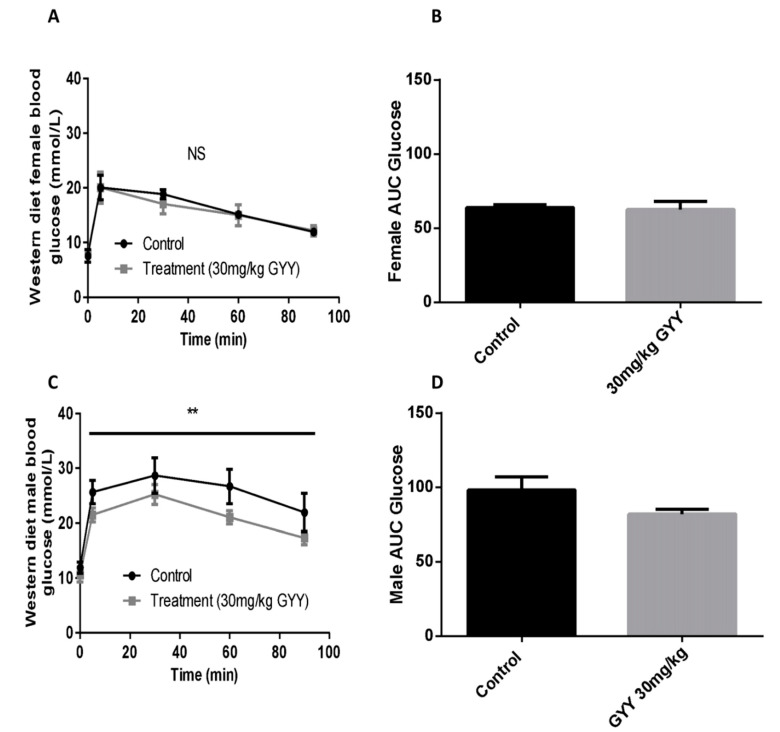
The glucoregulatory effect of H_2_S donor in Western diet-fed mice. Plasma glucose levels in response to an oral glucose challenge was examined in fasted mice given an intraperitoneal (IP) injection of GYY4137 (30 mg/kg) or saline at −16 h. Glucose concentration and area under the curve (AUC) were assessed in female (**A**,**B**) and male (**C**,**D**) mice. Results are expressed as absolute glucose means ± SEM over time after oral glucose challenge (2 g/kg). Data were analyzed using a Two-way ANOVA, ** = *p* < 0.05 effect for the treatment. N = 6 treatments and n = 5 controls.

**Figure 5 antioxidants-09-01038-f005:**
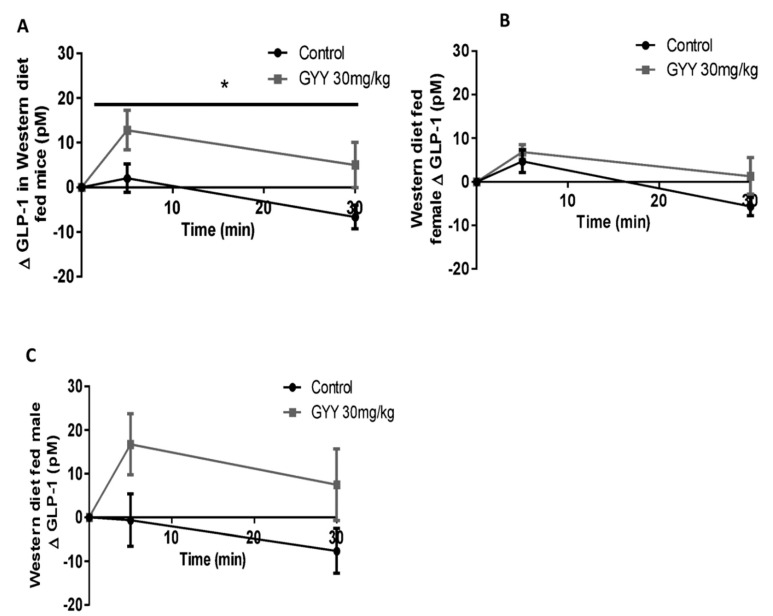
The effect of H_2_S donor on glucose-stimulated GLP-1 release in Western diet-fed mice. Plasma GLP-1 response to an oral glucose challenge was examined in fasted mice given an IP injection of GYY4137 (30 mg/kg) or saline at −16 h. GLP-1 response was assessed as combined male and female (**A**), female (**B**), and male mice (**C**). Results are expressed as the change in GLP-1 from baseline in means ± SEM over time after oral glucose challenge (2g/kg). Data were analyzed using a Two-way ANOVA, * = *p* < 0.05 effect for the treatment. n = 12 treatments and n = 10 controls in (**A**) and n = 6 treatments and n = 5 controls in (**B**,**C**).

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
