# Peer review of "The Protective Role of Hydrogen Sulfide Against Obesity-Associated Cellular Stress in Blood Glucose Regulation"

_antioxidants, 2020, doi:10.3390/antiox9111038_

Round 1

Reviewer 1 Report

Including the following information will increase quality of the work:

1.) Lines 135-136; 185-186; 218; 232; 281-282; 301+:  “Mice were injected intraperitoneally with 30 mg/kg GYY4137.”  

- Can you include in the discussion detailed information about the concentration of H2S released from GYY4137 throughout mice body for 16 hours?

- Was H2S concentration increased 16 hours after the injection?

- Can you include in the discussion the information about a life time of H2S in mice body and possible S-derivatives formed from H2S?

- Are reported effects of GYY4137 direct effect of H2S or its derivatives? H2S derivatives (e.g. polysulfides) were reported having higher antioxidant effects in comparison to H2S.

2.) Lines 45-46; 166-167; 252-253:

- Can you include detailed molecular mechanism of palmitic acid (PA, 16:0) causing metabolic stress leading to diabetes?

- Can you include molecular mechanism - how circulating palmitic acid (PA) increases concentration of cellular ROS?

- Molecular mechanism of impairment of the GLP-1?

Author Response

please see the attachement

Reviewer 2 Report

In the current manuscript by Mezouari et al., the ability of hydrogen sulfide (H2S) to protect against obesity-associated oxidative stress was explored. More specifically, the contributions of the incretin hormone, GLP-1, in this process was examined. Two different L cell lines and a mouse model of diet-induced obesity were used.

Concerns:

  1. ln 182-188- Why were NCI-H716 cells, and not GLUTag cells, used to explore the antioxidant effect of GYY4137? The effects of PA appear to be greater with respect to ROS production and cytotoxicity in GLUTag cells, as opposed to NCI-H716 cells (Fig 1.). To me, this would suggest that GLUTag cells would be a better suited model to study the effects of GYY4137?
  2. Figure 5: I am concerned by the way data are presented. Why are the male and female mouse data combined for Figure 5A? I suspect that significance was only reached in this case because the combination of male and female mice data reduced the variance. When plotted individually, no differences were observed (Figure 5B and C). As the results from figure 4 show that male mice respond better to GYY4137, the data should be shown separately.
  3. Figure 5: Why are the deltas being shown? I recognize that GLP-1 is difficult to measure in mice, but total levels should be measurable as long as the appropriate inhibitors are being included during sample collection. Data should be shown in absolute numbers.
  4. Ln 247: Please verify through the manuscript: Data is plural. Please correct phrases like “Data was shown.”

5.ln 326-329: It appears that these are instructions for how to write a conclusion. This should be deleted from the manuscript.
